# CTRP13-Mediated Effects on Endothelial Cell Function and Their Potential Role in Obesity

**DOI:** 10.3390/cells13151291

**Published:** 2024-07-31

**Authors:** Muhammad Aslam, Ling Li, Sina Nürnberger, Bernd Niemann, Susanne Rohrbach

**Affiliations:** 1Experimental Cardiology, Department of Internal Medicine I, Justus Liebig University Giessen, 35390 Giessen, Germany; muhammad.aslam@physiologie.med.uni-giessen.de; 2Institute of Physiology, Justus Liebig University Giessen, 35390 Giessen, Germany; ling.li@physiologie.med.uni-giessen.de (L.L.); sina.nuernberger@gmx.de (S.N.); 3Department of Cardiovascular Surgery Giessen, University-Hospital Giessen and Marburg, Justus Liebig University Giessen, 35390 Giessen, Germany; bernd.niemann@chiru.med.uni-giessen.de

**Keywords:** adipokines, endothelium, AMPK, proliferation, cell cycle, obesity

## Abstract

Background: Obesity, a major component of cardiometabolic syndrome, contributes to the imbalance between pro- and anti-atherosclerotic factors via dysregulation of adipocytokine secretion. Among these adipocytokines, the C1q/TNF-related proteins (CTRPs) play a role in the modulation of atherosclerosis development and progression. Here, we investigated the vascular effects of CTRP13. Results: CTRP13 is not only expressed in adipose tissue but also in vessels/endothelial cells (ECs) of mice, rats, and humans. Obese individuals (mice, rats, and humans) showed higher vascular CTRP13 expression. Human Umbilical Vein Endothelial Cells (HUVECs), cultured in the presence of serum from obese mice, mimicked this obesity-associated effect on CTRP13 protein expression. Similarly, high glucose conditions and TNF-alpha, but not insulin, resulted in a strong increase in CTRP13 in these cells. Recombinant CTRP13 induced a reduction in EC proliferation via AMPK. In addition, CTRP13 reduced cell cycle progression and increased p53 phosphorylation and p21 protein expression, but reduced Rb phosphorylation, with the effects largely depending on alpha-2 AMPK as suggested by adenoviral overexpression of dominant-negative (DN) or wild-type (WT) alpha 1/alpha 2 AMPK. Conclusion: The present study demonstrates that CTRP13 expression is induced in ECs under diabetic conditions and that CTRP13 possesses significant vaso-modulatory properties which may have an impact on vascular disease progression in patients.

## 1. Introduction

Metabolic syndrome (MetS), a cluster of comorbidities including visceral obesity, hypertension, insulin resistance, and dyslipidemia, constitutes a common risk factor for various cardiovascular diseases [1]. Obesity, in particular, is a predominant risk factor in developing insulin resistance or type 2 diabetes mellitus (T2DM) and is associated with higher cardiovascular mortality rates. Moreover, hyperglycemia and hyperinsulinemia not only contribute to cardiometabolic syndrome but also predispose to diabetic vasculopathy, including arteriosclerosis-related diseases such as coronary or peripheral artery disease (CAD and PAD, respectively). Adipose tissue (AT), long regarded as a simple energy storage organ, is now recognized as a dynamic endocrine organ. AT is involved in the regulation of various physiological functions through the secretion of so-called adipocytokines/adipokines. These secreted hormones and cytokines from AT play a crucial role in the communication between different organs and cells. Importantly, the secretory profile of AT differs significantly between obese and normal-weight patients. Among these adipocytokines, adiponectin is well-described and traditionally hailed as a protective factor with insulin-sensitizing, anti-inflammatory, anti-atherosclerotic, and cardioprotective effects [2]. Numerous studies have demonstrated an inverse correlation between plasma adiponectin levels and the incidence, severity, or outcome of CAD [3,4,5]. However, other studies failed to establish this correlation or even demonstrated a paradoxical link between higher adiponectin levels and negative events [6,7,8]. These conflicting data clearly highlight the necessity of understanding the involved bioactive mediators and their pathways for a better definition of potential therapeutic targets in MetS.

Members of the complement 1q (C1q)/tumor necrosis factor (TNF)-related protein (CTRP) family are secreted proteins composed of a C-terminal globular C1q (gC1q) domain and an N-terminal variable domain. The adipocytokine adiponectin is the best-studied representative of this family and an important regulator of glucose and lipid metabolism. In addition to adiponectin, there are 15 other structurally related ligands, from CTRP1 to CTRP15 [9]. Unlike adiponectin, which is primarily expressed in adipose tissue, most CTRPs exhibit a broader pattern of expression. The present study tried to elucidate the potential role of CTRP13, one member of the large CTRP protein family, in obesity- or T2DM-induced endothelial dysfunction. CTRP13 is highly expressed in both adipose tissue and the brain [9]. AMPK and Akt activation seem to be the central signaling events that mediate the metabolic effects of various CTRPs, including CTRP13 [9,10,11].

The CTRPs, including CTRP13, have been associated with various cardiometabolic effects. Initially, CTRP13 demonstrated the ability to reduce food intake and body weight in mice when delivered centrally [12]. However, more recently, the same group demonstrated that CTRP13 is a negative metabolic regulator, and its deficiency improves glucose and lipid handling, elevates physical activity, and lowers body weight [13]. CTRP13 has been found to play a role in pathophysiological mechanisms that influence the progression of coronary artery disease (CAD) progression by regulating endothelial function, inflammatory response, and metabolism [14]. While increasing serum levels of CTRP1 and CTRP5 were associated with the severity of CAD, CTRP13, on the contrary, was reported as a protective factor in CAD [14]. Infusion of recombinant CTRP13 in ApoE KO mice has been described to reduce atherosclerotic lesions by modulating lipid uptake and foam-cell formation [15]. Additionally, CTRP13 was shown to preserve endothelial function in diabetic mice by regulating endothelial nitric oxide synthase (eNOS) coupling [16], preventing vascular calcification through destabilization and degradation of the master transcription factor for osteogenesis Runx2 [17], and reducing the incidence and severity of abdominal aortic aneurysm (AAA) through reduced aortic macrophage infiltration [18]. Here, we analyzed the impact of obesity on vascular CTRP13 expression in three different species (mouse, rat, and human) and investigated potential mechanisms involved in obesity-associated increased CTRP13 expression in endothelial cells (ECs). These analyses were complemented by studies on the influence of CTRP13 on EC function and signal transduction.

## 2. Materials and Methods

### 2.1. Mouse Model

Age-matched, female and male ob/ob mice (n = 6 per group) on a C57BL/6J background (The Jackson Laboratory, Bar Harbor, ME, USA) and the according littermates were housed in an environmentally controlled laboratory with a 12 h light–dark cycle accommodating 2–4 mice in each cage. Mice were sacrificed by cervical dislocation at the age of 6 months. Fasting serum parameters were measured after a 12 h food withdrawal prior to euthanasia. Organs were removed, dissected, and unless otherwise specified, stored in liquid nitrogen. Blood was drawn by aortic puncture at the time of sacrifice. All animal experiments conformed to the guidelines outlined in Directive 2010/63/EU of the European Parliament regarding the protection of animals used for scientific purposes. The experiments were performed in accordance with the regional authorities and ethics committees for animal research (AZ 468_M).

### 2.2. Rat Model

Obesity-associated changes in CTRP13 expression were analyzed in male obese ZDF (fa/fa) rats, lean heterozygous (Fa/fa) rats, and wild-type (Fa/Fa) rats (Charles River Laboratories). In the ZDF rat with a loss-of-function mutation in their leptin receptors, the onset of obesity occurs at 4 weeks of age, while at 12 weeks of age, overt diabetes begins. In this study, animals at 16 weeks of age were used. All animal experiments conformed to the guidelines outlined in Directive 2010/63/EU of the European Parliament regarding the protection of animals used for scientific purposes. The experiments were performed in accordance with the regional authorities and ethics committees for animal research (AZ 596_M).

### 2.3. Patient Samples

The remainder of internal mammary artery tissue, not utilized for coronary artery bypass grafting (CABG), was obtained from 30 patients undergoing cardiac surgery. The subjects were grouped according to their BMI (18.5–25 kg/m^2^: normal weight or 30–35 kg/m^2^: obese). Patients with pre-existing type 2 diabetes mellitus (T2DM) (fasting glucose > 120 mg/dL; on diabetes medications), reduced left ventricular (LV) function (ejection fraction (EF) < 55%), or a BMI between 25.1 and 29.9 kg/m^2^ were excluded from study participation. Approval for the use of human tissue was granted by the local ethics committee (AZ 65/10 and AZ 229/18) and informed patient consent was obtained before surgery.

### 2.4. Cell Isolation and Culture

Human umbilical vein endothelial cells (HUVECs) were isolated from umbilical cords obtained from the gynecology department at University Hospital Giessen after approval from the ethics committee of the hospital and informed consent was obtained from the patients. The isolated cells were cultured in endothelial cell basal medium (PromoCell GmbH, Heidelberg, Germany) supplemented with 10% (*v*/*v*) FCS, 0.4% (*v*/*v*) endothelial growth supplement (PromoCell, GmbH, Heidelberg, Germany) with heparin, 0.1 ng/mL human EGF, 1.0 μg/mL hydrocortisone, 1 ng/mL human bFGF and 2% (*v*/*v*) penicillin/streptomycin in humidified atmosphere at 37 °C, 5% CO_2_. Confluent cultures of primary ECs were trypsinized in phosphate buffer saline (PBS), supplemented with 0.05% (*w*/*v*) trypsin, and 0.02% (*w*/*v*) EDTA, and seeded at a density of 7 × 10^4^ cells/cm^2^ on 35-mm culture dishes. Experiments were performed 2–3 days after seeding using confluent endothelial monolayers of passage 1 grown in Opti-MEM™ supplemented with 1% serum unless otherwise stated. The study was approved by the local institutional Ethics Committee (AZ 18/13).

To isolate human primary cardiac fibroblasts, right atrial (RA) tissue from patients undergoing cardiac surgery was collected in ice-cold phosphate-buffered saline (PBS). The tissue was trimmed of excess fat, and cardiac trabeculae were isolated. Using scissors, cardiac trabeculae were cut into small pieces and then minced into 1–2 mm pieces with a tissue chopper (McIlwain, The Mickle Laboratory Engineering Co. Ltd., Surrey, UK). The cardiac tissue was enzymatically disaggregated by incubation in 0.25% trypsin and 0.1% (*w*/*v*) collagenase II (Sigma-Aldrich, Taufkirchen, Germany) in Krebs–Henseleit buffer for 60 min at 37 °C under gentle agitation and constant gasification with carbogen (95% O_2_/5% CO_2_). The cell suspension was then removed from the minced tissue by passage through a 100 μm cell strainer. To sediment the cells, the suspension was centrifuged for 10 min at 300× *g*. The pellet was resuspended in DMEM supplemented with L-glutamine, penicillin/streptomycin, and 10% (*v*/*v*) FCS. The cells were seeded onto 15 cm culture dishes and maintained at 37 °C and 5% CO_2_ in a humidified incubator. After incubation for 3 h at 37 °C, non-adherent cells were removed by rinsing with PBS. When confluent, cells were passaged using a trypsin/0.02% EDTA solution and seeded at densities of 10,000 cells/cm^2^. The study was approved by the local institutional Ethics Committee (AZ 33/16).

### 2.5. RNA Isolation and qPCR

Total RNA was isolated from mouse tissue and HUVECs using TRIzol™ Reagent (Thermo Fisher Scientific, Darmstadt, Germany) in accordance with the manufacturer’s instructions. The integrity and quality of the RNA was determined by gel electrophoresis. Prior to cDNA synthesis, the RNA concentration was determined by measuring UV absorption. Reverse transcription of RNA samples (500 ng total RNA) was carried out for 30 min at 42 °C using the SuperScript™ III First-Strand cDNA Synthesis Kit (Thermo Fisher Scientific, Darmstadt, Germany). Real-time PCR and subsequent data analysis were performed using the qTOWER^3^ G thermocycler (Analytik Jena AG, Jena, Germany). Each assay was performed in duplicate, and the validation of the PCR runs was assessed by evaluation of the melting curve of the PCR products (primer sequences in Appendix A). Threshold cycles (CTs) of target genes were normalized to the mean of 18S rRNA, HPRT1, and GAPDH. The obtained ΔCT values were compared to controls and relative mRNA expression was calculated using the formula R = 2^(−ΔΔCT)^.

### 2.6. Western Blotting

Tissues or cells were homogenized in a buffer containing 50 mmol/L Tris HCl, 150 mmol/L NaCl, 5 mmol/L EDTA, 1% SDS, 1% sodium deoxycholate, and protease and phosphatase inhibitor cocktails (Sigma-Aldrich Chemie GmbH, Schnelldorf, Germany). Subsequently, 20–50 µg of protein was loaded on an SDS-PAGE gel and transferred to a nitrocellulose membrane. Following blocking, filters were exposed to primary antibodies at 4 °C overnight. After incubation with a peroxidase-conjugated secondary antibody, blots were subjected to the enhanced chemiluminescent detection method with the Fusion FX7 imaging system (Peqlab Biotechnologie GmbH, Erlangen, Germany).

### 2.7. Isolation of Recombinant CTRP13

The recombinant protein was produced by cloning full-length mouse CTRP13 into pENTR™/D-TOPO^®^ (Thermo Fisher Scientific, Darmstadt, Germany) and maintained in *E. coli*. strain TOP10. After recombination into pDEST17, the N-terminal His6-tagged fusion protein was produced in *E. coli* strain BL21-AI. The protein was isolated from the lysed bacterial pellet with a nickel-affinity column (Amocol Bioprocedures Limited, Teltow, Germany), eluted with imidazole-containing buffer, and dialyzed against phosphate-buffered saline (PBS). Potential endotoxin contaminants were removed with the EndoTrap Red Kit (Hyglos GmbH, Bernried, Germany). The absence of endotoxin was verified with the Pierce Chromogenic Endotoxin Quant Kit (Thermo Fisher Scientific, Darmstadt, Germany) with a lower detection limit of 0.01 EU/mL. Control cells were treated with equal amounts of recombinant GFP (green fluorescent protein) isolated under identical conditions.

### 2.8. Immunocytochemistry and Confocal Microscopy

HUVECs were cultured until confluence on glass cover slips. After treatment, cells were fixed with 4% PFA, permeabilized with 0.2% Triton X-100, and blocked with blocking solution (5% BSA + 5% FCS) for 1 h. Cells were incubated with the primary CTRP13 antibody overnight at 4 °C, washed with PBS, and afterward stained with the secondary, Cy3-labeled antibody for 1 h at room temperature (RT). The ECs were also stained with an anti-CD31 antibody for 3h at RT, followed by washing and a secondary, FITC-conjugated antibody. Nuclei were labeled with TO-PRO™-3 Iodide (Thermo Fisher Scientific, Darmstadt, Germany). Finally, the cover slips were embedded in a fluorescent mounting medium (CitiFluor, Hatfield, PA, USA) and placed onto glass slides. Images were obtained using a Zeiss LSM 710 (Zeiss; Jena, Germany) confocal microscope.

### 2.9. Serum Analyses

Serum TNF-alpha and insulin in mice and rats were measured by using commercial Enzyme-Linked-Immunosorbent-Assays (Mouse TNF-alpha ELISA Kit, Raybiotech; Mouse Insulin ELISA, Mercodia AB; Rat TNF-alpha ELISA Kit, Raybiotech; Rat Insulin ELISA, Mercodia AB). Serum glucose values were determined using a Glucose Assay Kit (BioCat GmbH, Heidelberg, Germany).

### 2.10. Migration Assay

To assess cell migration, silicone culture inserts (Ibidi GmbH, Graefelfing, Germany) were utilized, which enabled a defined cell-free gap with a width of 500 μm. A total of 20,000 HUVECs or RA fibroblasts were seeded into the culture insert and cultured until a confluent monolayer formed. After the removal of the silicone insert, cells were cultured in Opti-MEM medium with 1% FBS. To investigate the effect of CTRP13 on EC migration, monitoring was performed with the JuLI™ Br Live Cell Analyzer (Peqlab Biotechnologie GmbH, Erlangen, Germany).

### 2.11. Sprouting Assay

The cell sprouting assay was performed as described by our group previously [19]. Briefly, HUVECs were grown to 70–75% confluence, trypsinized, and mixed with endothelial growth medium (EGM) as described above with 20% FCS and 0.2% methyl cellulose. The cell mixture was then incubated overnight at 37 °C in uncoated, round bottom, sterile 96-well culture dishes. The cells formed spheres around the methyl cellulose particles. These spheres were collected via centrifugation at 800 RPM and resuspended in EGM. Equal volumes of cell sphere suspension and rat tail collagen I solution (2 mg/mL) plus 1% NaOH solution (10 µL) were then combined and plated in 24-well tissue culture plates. The plates were incubated at 37 °C for 30 min, overlaid with EGM containing CTRP13 (4 μg/mL), VEGF (25 ng/mL) or VEGF + CTRP13 at increasing concentration (4, 8, 12 μg/mL), and then incubated at 37 °C overnight in a cell culture incubator. After 24 h, cell sprouts were photographed, and sprout length was analyzed.

### 2.12. Cell Cycle Analysis

Following a 24 h incubation with CTRP13 (4 µg/mL), HUVECs were harvested, trypsinized, and adjusted to a concentration of 1 × 10^6^ cells/mL. The cells were fixed in 75% ethanol at −20 °C, treated with RNase A (10 mg/mL) for 1 h at 37 °C, stained with propidium iodide (PI) (50 μg/mL), and subjected to DNA content analyses using fluorescence-activated cell sorting (FACS) on a FACS Calibur, (Becton-Dickinson, Heidelberg, Germany). The analysis was performed with CellQuest Software version 3.1 (BD Bio-sciences, San Jose, CA, USA).

### 2.13. Adenovirus Generation and Utilization

The human dominant negative (DN) construct of alpha 1 or alpha 2 AMPK encodes AMPK mutated at position 172 to an alanine (T172A), as described previously [20]. Both mutations were introduced by site-directed mutagenesis of wild-type (WT) alpha 1 and alpha 2 AMPK using the Q5^®^ Site-Directed Mutagenesis Kit (NEB Inc., Ipswich, MA, USA). The constructs were cloned into pENTR™/D-TOPO™ (ThermoFisher Scientific, Waltham, MA, USA) and were confirmed by sequencing at Eurofins Genomics. Subsequently, the pENTR constructs were recombined using LR Clonase II into the expression vector pAd/CMV/V5-DEST™ Gateway^®^ Vector (ThermoFisher Scientific). The resulting expression constructs were transfected into the 293A cell line (ThermoFisher Scientific) to generate a crude adenoviral stock. After further amplification, the resulting viral stock was employed to transduce HUVECs. In addition, CTRP13 and GFP from pENTR™/D-TOPO^®^ were also recombined into pAd/CMV/V5-DEST™, and viral stocks were generated as described above. HUVECs were thereafter transduced with CTRP13, GFP, or treated with medium from 293A cells (controls). Then, 48 h after transduction, HUVECs were washed thoroughly to remove any remaining adenoviral particles and supplied with fresh EGM. Conditioned medium from HUVECs overexpressing either CTRP13 or GFP or HUVECs treated with conditioned medium from 293A cells was utilized to culture human fibroblasts in 50% (*v*/*v*) conditioned medium in DMEM.

### 2.14. BrdU Incorporation

The BrdU Cell Proliferation Assay Kit (Biovision) was used following the manufacturer’s instructions. Briefly, HUVECs were seeded at a density of 5000 cells per well in a 96-well microplate. After 24 h, BrdU was added to the cells for 6 h. Subsequently, cells were washed with PBS and fixed with paraformaldehyde (3.7%) for 15 min at 37 °C, and permeabilized with 0.2% Triton-X100 for 20 min. Subsequently, hydrochloric acid was used as follows: 1 N HCl for 10 min, then 2 N HCl for 10 min. Finally, a phosphate-citrate buffer was added and incubated for 10 min. After a washing step with PBS, the anti-BrdU antibody was added and incubated for 1.5 h at RT. Subsequently, the solution was removed and the secondary anti-mouse IgG-HRP-coupled antibody was added for 1 h at RT. After washing, the TMB substrate solution supplied in the kit was added. Before reading at a wavelength of 450 nm on the microplate reader (Infinite 200 Pro, Tecan Deutschland GmbH, Crailsheim, Germany), the TMB reaction was stopped.

### 2.15. Antibodies and Chemicals

CTRP13 (Biorbyt, Cambridge, UK # orb155966), alpha-Tubulin (Cell Signaling, Danvers, MA, USA # 2144), GAPDH (Cell Signaling #2118), p53 (Cell Signaling #9282), phospho-p53 (Ser20; Cell Signaling # 9287), phospho-p53 (Ser15, Cell Signaling # 9284), p21 (Cell Signaling # 2947), phospho-Rb (Ser795, Cell Signaling # 9301), CD31 (AgilentDako, Santa Clara, CA, USA # M0823), p44/42 MAPK (Cell Signaling #9102), phospho-p44/42 MAPK (Cell Signaling #9101), AMPK (Cell Signaling #2532), phospho-AMPK (Thr172, Cell Signaling #2535), Akt (Cell Signaling #9272), phospho-Akt (Thr308, Cell Signaling #4056), V5 (ThermoFisher # R960-25), Smad2/3 (Cell Signaling # 5678), phospho-Smad2/3 (Cell Signaling # 8828), adenine 9-β-D-arabinofuranoside (Jena Bioscience, Jena Germany # NU-875), SB202190 (Sigma-Aldrich, Taufkirchen, Germany # 559388), U0126 (Sigma-Aldrich # U120), human TNF-alpha (Sigma-Aldrich # H8916), C87 (Bio-Techne, Wiesbaden, Germany # 5484), and VEGF (Miltenyi Biotec, Bergisch Gladbach, Germany # 130-127-426).

### 2.16. Statistical Analysis

All data are presented as mean ± SEM. Statistical analyses were performed using SigmaStat 3.5 software (Systat Software, Inc., Palo Alto, CA, USA). The Data were analyzed for normal distribution (Shapiro–Wilk test) and variance (Levene test) and subsequently analyzed using Student’s t test or ANOVA with post hoc analysis as appropriate. A significance level of *p* < 0.05 was considered statistically significant.

## 3. Results

### 3.1. Impact of Obesity on CTRP13 Expression

#### 3.1.1. Impact of Obesity on CTRP13 Expression in Mice and Rats

CTRP13 is preferentially expressed in adipose tissue and brain of mice [21]. Additionally, we demonstrate significant expression of CTRP13 mRNA (Appendix A) and protein (Appendix A) in the mouse aorta and heart. Ob/ob mice showed a strong increase in body weight, fasting glucose, and plasma insulin compared to their wild-type littermates at 6 months of age (Table 1), indicating the presence of T2DM in these mice.

In accordance with the previous reports [21], the expression of visceral adipose tissue CTRP13 mRNA and protein increased in leptin-deficient obese (ob/ob) hyperphagic mice (Figure 1A). In addition to the expression changes in adipose tissue, aortic tissue from these obese mice demonstrated a similar increase in CTRP13 mRNA and protein expression (Figure 1B) compared to lean, wild-type mice. Although an obesity-associated increase in CTRP13 expression was observed in adipose and aortic tissue of both male and female mice, the increase in expression was significantly greater in males (Figure 1A,B). The obesity-associated increase in aortic CTRP13 expression was not restricted to ob/ob mice but was also observed in obese ZDF (fa/fa) rats, a widely used rat model of genetic obesity and T2DM with a loss-of-function mutation in their leptin receptors (Figure 1C). The obese ZDF (fa/fa) rats showed a significant increase in body weight at the age of 16 weeks compared to lean heterozygous (Fa/fa) and wild-type (Fa/Fa) rats (Table 2). Additionally, fa/fa rats exhibited higher fasting glucose and insulin plasma levels indicating the presence of T2DM (Table 2). Heterozygous (Fa/fa) rats did not present higher body weights but moderately increased glucose and insulin levels, indicating impaired glucose tolerance (Table 2). Compared to wild-type (Fa/Fa) rats, lean heterozygous (Fa/fa) and obese, diabetic, homozygous (fa/fa) rats showed a significant increase in CTRP13 mRNA and protein expression in aortic tissue (Figure 1C). A similar phenomenon was also observed in their adipose tissue and cardiac tissue (Figure 1D). Within the heart, CTRP13 mRNA is highly expressed in cardiac microvascular ECs with significantly lower expression levels in adult rat cardiomyocytes or cardiac fibroblasts (Appendix A). These data suggest that not only circulating CTRP13 from adipose tissue but also locally produced CTRP13 in ECs may mediate paracrine vascular effects in both mice and rats.

#### 3.1.2. Impact of Obesity on CTRP13 in Patient Samples

Following up on the observed high endothelial expression of CTRP13 in rodents, we next investigated whether CTRP13 is also modulated by obesity in patient tissue. For this purpose, we analyzed the remainder of internal mammary artery tissue that was not utilized for coronary artery bypass grafting (CABG). Mammary arteries from obese patients (BMI: 30–35 kg/m^2^) exhibited a significant elevation in both CTRP13 mRNA and protein levels compared to arteries from lean patients (BMI: 18.5–25 kg/m^2^), as shown in Figure 2. Notably, no significant differences were observed between male and female patients in our cohort.

Apart from differences in BMI, these two patient groups did not differ in parameters such as age, left ventricular ejection fraction, surgical characteristics (number of grafts, aortic cross-clamping time, or cardiopulmonary bypass time), inflammatory markers (preoperative CRP or leucocytes) or medication (Table 3). Moreover, there was no significant difference in HbA1c between lean and obese patients (Table 3). However, fasting glucose levels were moderately elevated in obese patients (Table 3) despite adherence to our exclusion criteria (pre-existing diabetes mellitus; fasting glucose > 120 mg/dL; diabetes medications). Collectively, the data gathered from mice, rats, and humans suggest that obesity significantly contributes to altered CTRP13 homeostasis across all three species.

### 3.2. Mechanisms Involved in Obesity-Associated CTRP13 Induction in Ecs

In the rat heart, CTRP13 mRNA is highly expressed in cardiac microvascular endothelial cells (Appendix A). Similarly, human ECs (HUVECs) exhibit cytosolic expression of CTRP13 (Figure 3A). Under baseline conditions, CTRP13 protein expression is minimal; however, it can be significantly induced when cells are cultured in the presence of serum from ob/ob mice (Figure 3B), suggesting that the blood of these mice contains CTRP13-inducing mediators.

Plasma glucose and insulin were elevated in both obese mice and rats (Table 1 and Table 2) with moderately higher fasting glucose observed in obese patients as well (Table 3). In addition, plasma TNF-alpha, which showed significantly higher levels in type 2 diabetes mellitus (T2DM) mice and rats (Table 1 and Table 2), plays a crucial role in inducing insulin resistance [22], a circumstance that is also known from T2DM patients [23]. Therefore, we tested the hypothesis of whether insulin, glucose, or TNF-alpha contribute to the elevated CTRP13 expression in ECs. Interestingly, HUVECs treated with increasing concentrations of insulin (1–1000 nM) for 24 h did not demonstrate an increased expression of CTRP13. Conversely, cultivating the cells under high glucose conditions (25 mM D-glucose) induced upregulation of CTRP13 protein expression in HUVECs (Figure 4A) compared to cells maintained under normal glucose levels (5 mM D-glucose) for 24 h. Osmolarity controls (25 mM L-glucose + 5 mM D-glucose) did not show an increased CTRP13 expression. Likewise, treatment of HUVECs with TNF-alpha (1 ng/mL or 10 ng/mL) under normal glucose conditions for 24 h resulted in increased CTRP13 protein expression (Figure 4B). The combination of both conditions (high glucose + low conc. TNF-alpha) seemed to have an additive effect on CTRP13 protein expression (Figure 4C). A specific small molecule inhibitor of TNF-alpha, C87, [24] efficiently blocked the TNF-alpha-mediated increase in CTRP13 protein expression, suggesting an essential role of the cytokine in this process (Figure 4D). C87 directly binds to TNF-alpha and effectively blocks TNF-alpha-triggered signaling and gene expression, including its own expression as shown by others [24]. Therefore, we tested whether C87 also impacts high glucose-induced CTRP13 expression. The experiments showed (Figure 4E) that C87 reduced CTRP13 protein expression in cells cultured under high glucose conditions. Furthermore, C87 treatment also resulted in diminished mRNA of NF-KB target genes such as ICAM-1, IL-8, and TNF-alpha itself under high glucose conditions (Figure 4F), suggesting that TNF-alpha-triggered signaling is indeed involved in increased CTRP13 expression under these conditions.

### 3.3. Functional Impact of CTRP13 on Endothelial Cells

#### 3.3.1. Signal Transduction Induced by CTRP13

Subsequently, we aimed to elucidate the functional consequences of an increased release of CTRP13 on ECs, considering that, unlike adiponectin, most CTRPs do not circulate at high concentrations systemically but may instead mediate local, paracrine, or autocrine effects. HUVECs were treated with 4 µg/mL of recombinant CTRP13, a concentration within the range previously employed for CTRP13 by others [21] and us [10] in various cell types. CTRP13 induced a rapid (within 10 min) and sustained (24 h) activation of AMP-activated protein kinase (AMPK) and a transient activation of p44/42 MAPKinase (Figure 5). Consistent with the effects on AMPK activation, phosphorylation of protein kinase B (Akt) at Thr 308 was evident in response to CTRP13 after 20 min and persisted up to 24 h (Figure 5). Notably, only minor and transient effects of CTRP13 on Akt phosphorylation at Ser 473 were detectable (Figure 5).

#### 3.3.2. EC Proliferation and Migration

Next, we investigated the impact of CTRP13 on the proliferation and migration of HUVECs. Cultivation of ECs in the presence of CTRP13 resulted in a significant reduction in cell number compared to the controls (Figure 6A). Notably, the inhibition of AMPK with adenine 9-β-D-arabinofuranoside (AraA) largely mitigated these effects of CTRP13, whereas inhibition of p44/42 MAPKinase with UO126 or the use of the Akt inhibitor VIII had no significant impact on the effects of CTRP13 on cell number (Figure 6A). Similarly, CTRP13 significantly decreased BrdU incorporation in ECs, indicating reduced cell proliferation. This effect was inhibited by AraA but not by UO126 or Akt inhibitor VIII (Figure 6A). Moreover, the scratch assay used to assess cell migration revealed a significant reduction induced by CTRP13 in an AMPK-dependent manner (Figure 6B). Collectively, these data suggest that CTRP13-dependent reduction in EC migration and proliferation is largely mediated via AMPK. Finally, we performed sprouting assays to test the angiogenic potential of CTRP13. Vascular endothelial growth factor (VEGF) is known to induce in vitro EC sprouting [25] and was therefore used as a positive control. As shown in Appendix A, the impact of CTRP13 on EC sprouting was very low. Only the combination with VEGF showed a minor increase in sprout length, suggesting a moderate additive effect of CTRP13. In addition, we also performed sprouting assays in CTRP13-overexpressing HUVECs in the absence or presence of TNF-alpha to mimic an obese environment. However, no major differences could be detected between the groups.

#### 3.3.3. EC Cell Cycle Progression and Role of AMPK Isoforms

To further understand the mechanism underlying the effects of CTRP13 on EC migration and proliferation, additional experiments were conducted focusing on three known negative regulators of cell cycle progression: p53, p21, and retinoblastoma protein (Rb). Activation of AMPK has been previously linked to the phosphorylation of p53 at Ser15, an event necessary to initiate AMPK-dependent cell-cycle arrest [26]. Additionally, Ser15 phosphorylation triggers additional phosphorylation events in p53, including phosphorylation of Ser20 or Thr18, which contribute further to p53 induction and activation [27]. Upon treatment with CTRP13, an increased phosphorylation of p53 at Ser15 but not at Ser20 was observed. Additionally, an increased p21 protein expression, a downstream target of p53 that prevents the cell from transitioning from the G1 to the S phase of the cell cycle, was observed following CTRP13 treatment (Figure 7A). Moreover, Rb phosphorylation decreased in response to treatment of CTRP13 (Figure 7A), indicating Rb activation and consequent inhibition of cell cycle progression through binding to transcription factors, preventing them from initiating gene transcription. All these effects of CTRP13 were nearly completely prevented by AMPK inhibition (AraA). In addition, FACS-based cell cycle analyses suggested that treatment of CTRP13 results in a higher number of ECs in G1/G0, inhibiting progression into the S-phase of the cell cycle (Figure 7A) with no observable impact on the G2/M phase (Figure 7A).

HUVECs express both isoforms of AMPK, but alpha 1 AMPK appears to be the dominant isoform in this cell type (Appendix A). To determine which AMPK isoform is crucial for the observed CTRP13 effects on the cell cycle, adenoviral overexpression of wild-type (WT) or dominant negative (DN) alpha 1 and alpha 2 AMPK in HUVECs was performed. As shown in Figure 7B, DN alpha 2 AMPK but not DN alpha 1 AMPK overexpression, blunted p53 phosphorylation (Ser 15), blunted increased p21 expression and inhibited p-RB de-phosphorylation in response to CTRP13, suggesting that alpha 2 AMPK is primarily responsible for these effects. Consistent with this, the FACS analyses revealed that the shift towards G1/G0 and the inhibited progression towards S-phase were blunted after DN alpha 2 AMPK but not DN alpha 1 AMPK overexpression (Figure 7B). No major impact on the G2/M phase was observed (Appendix A).

### 3.4. Functional Impact of Endothelial CTRP13 on Human Fibroblasts

Next, we tested the impact of endothelial CTRP13 on neighboring cells. Conditioned medium from CTRP13-overexpressing HUVECs (Figure 8A), containing large amounts of secreted CTRP13 (Figure 8A), was utilized to treat human fibroblasts. Unlike ECs, human fibroblasts showed very low endogenous expression of CTRP13 mRNA or protein (Appendix A). As shown in Figure 8B, treatment of fibroblasts with conditioned medium from CTRP13-overexpressing HUVECs resulted in a moderately increased p53 phosphorylation at Ser15, increased p21 protein expression, and increased alpha-AMPK phosphorylation at Thr172. These effects on fibroblasts were comparable to those observed in HUVECs (Figure 5 and Figure 7A). In addition, a reduction in the phosphorylation of Smad2 and Smad3, major regulators of the fibrogenic transcriptional program, could be demonstrated in fibroblasts (Figure 8C). Accordingly, conditioned medium from CTRP13-overexpressing cells induced a significant reduction in the expression of the fibrosis- or myofibroblast phenotype-related genes such as collagen I (Col1A1), collagen III (Col3A1), Transforming Growth Factor-beta1 (TGF-beta1), or alpha-smooth muscle actin (alpha-SMA) in human fibroblasts (Figure 8D), and a significant deterioration of their migratory potential compared to fibroblasts treated with conditioned medium from GFP-overexpressing HUVECs (Figure 8E).

## 4. Discussion

The present study demonstrates that CTRP13 expression is induced under diabetic conditions in ECs in vitro and in vessels in vivo. The induction of vascular CTRP13 occurs in rodent models of obesity as well as in obese patients, suggesting a universal mechanism. High glucose conditions and exposure to TNF-alpha, which is known to play a crucial role in inducing insulin resistance and T2DM, are strong inducers of endothelial CTRP13 expression in vitro. Functional analyses reveal that CTRP13 inhibits proliferation, migration, and cell cycle progression in ECs in an AMPK-dependent manner but does not seem to have a major impact on angiogenesis processes. In addition, endothelial-derived CTRP13 induces signaling pathway activation in human fibroblasts, suggesting anti-proliferative and anti-fibrotic effects in these cells in vitro. This could be in accordance with the paracrine effects of endothelial CTRP13 with a pathophysiologically relevant influence on neighboring cells such as fibroblasts or smooth muscle cells in vivo.

Endothelial dysfunction results from an imbalance in the production of vasodilatory and vasoconstricting factors and predisposes the endothelium toward a prothrombotic and proatherogenic state. Obesity exacerbates this imbalance via dysregulation of adipokine synthesis through an increased release of proinflammatory adipokines, thus linking inflammatory milieu and vascular pathology. In obesity-related endothelial dysfunction, several factors come into play, including endothelial barrier disruption, the presence of oxidized low-density lipoprotein (oxLDL), diminished nitric oxide (NO) bioavailability and eNOS uncoupling, and increased reactive oxygen species (ROS) production. Additionally, increased macrophage infiltration contributes to adipose tissue-related inflammation. The impact of cardiometabolic syndrome, characterized by obesity, insulin resistance, and DM, on atherosclerosis is well-documented due to its strong association with macrovascular complications, particularly CAD [28,29]. Despite the diverse conditions leading to increased expression of the two adipokines CTRP13 and adiponectin, seemingly both exert similar effects on vascular function. The vasoprotective effects of adiponectin involve increased EC proliferation and migration, NO release [30], inhibition of atherogenic factors such ICAM-1 or E-Selectin [3], reduced monocyte adhesion [31], or priming of monocyte differentiation into anti-inflammatory M2 macrophages [32], and thus intervening in early arteriosclerosis events. Although experimental evidence is not as detailed as for adiponectin, CTRP13 has also been suggested to mediate comparable anti-atherosclerotic effects. These include modulation of lipid uptake and foam-cell formation [15], regulation of eNOS coupling [16], amelioration of oxLDL-induced loss of EC integrity [15], and reduction in macrophage infiltration [18].

CTRP13 is one of the highly conserved members of the C1q/TNF family [16,21] from zebrafish to humans, which suggests a potentially highly conserved biological function. Despite the similar expression profiles, characterized by strong expression in adipose tissue, and comparable structural features, CTRP13 and adiponectin exhibit only limited amino acid identity (39%) [21]. This suggests that, despite certain similarities, the two proteins have distinct molecular compositions, potentially contributing to differences in their functional roles and/or regulatory mechanisms. The plasma concentration of adiponectin is very high, reaching up to 20 μg/mL, which is over 1000 times higher than the concentrations of other cytokines [33]. In contrast, reported circulating CTRP13 serum levels have been shown to be below 5 ng/mL [34,35]. Consequently, circulating CTRP13 levels are significantly lower than adiponectin levels, suggesting that local CTRP13 effects, induced through autocrine or paracrine mechanisms, may be more important than its systemic effects in comparison to adiponectin.

There is an ongoing controversy regarding the altered CTRP13 expression and its role in obesity or T2DM. Our study adds to this discourse by providing evidence of increased vascular expression of CTRP13 in both male and female mice, rats, and humans. Elevated levels of CTRP13 transcript and circulating levels have been described in male obese mice [12,21], while others have reported significantly lower levels of CTRP13 in T2D patients [35] and in obese children [36]. Interestingly, no increase in serum CTRP13 has been observed in female obese mice [21] or overweight/obese premenopausal women compared to normal-weight controls [37]. Adiponectin expression or release, on the contrary, is reduced in obese animals and humans [38,39]. Interestingly, our study reveals that TNF-alpha treatment leads to an increased CTRP13 expression in ECs, which differs from the suppressive effects of TNFα treatment on adiponectin gene expression [40,41]. In vivo, TNF-alpha infusion decreases insulin sensitivity, causing a significant decrease in adiponectin in rats [41]. TNF-alpha is highly expressed in adipose tissues of obese humans and animals [42], inducing various characteristics of insulin resistance in experimental animals [43]. Notably, the loss of TNF-alpha has been linked to improved insulin sensitivity and protection against obesity-related reduction in insulin receptor signaling in obese mice [44]. Therefore, TNF-alpha is recognized as an important mediator linking obesity to insulin resistance. By contrast, insulin, a known enhancer of adiponectin expression and secretion in adipocytes [40], did not impact endothelial CTRP13 expression in the present study. In addition to TNF-alpha, high glucose also induced endothelial CTRP13 expression, and these effects appeared to involve TNF-alpha-induced signal transduction and gene expression as well. It is worth noting that others have reported CTRP13 downregulation under high glucose conditions in vivo and in vitro [45]. However, lentiviral-mediated ectopic overexpression of CTRP13 under high glucose conditions contributes to the reversal of pathological changes typically observed during hepatic fibrogenesis [45]. These findings underscore the intricate and context-dependent regulatory mechanisms governing CTRP13 expression in response to different stimuli. Initially, treatment with recombinant CTRP13 demonstrated the ability to reduce food intake and body weight in mice, as well as promote glucose uptake and improve insulin sensitivity in vitro [12,21]. However, more recent data has suggested a contrasting role of CTRP13 as a negative metabolic regulator, and loss-of-function models (CTRP13 KO mouse) have shown improved glucose tolerance, enhanced insulin sensitivity, and reduced inflammation in adipose tissue [13]. The discrepancy between the gain-of-function and loss-of-function studies raises questions about how accurately the use of recombinant protein reflects the physiological effects of CTRP13.

In addition to the functional and structural similarities, CTRP13 and adiponectin potentially also share signal transduction. Adiponectin mediates its metabolic effects mostly via its receptors, AdipoR1 and AdipoR2 [46]. Given the structural similarities, it is plausible that these receptors may also mediate the CTRP effects. Indeed, AdipoR1 was identified as a putative receptor for CTRP7 [10] and CTRP9 [9,11,47]. Nevertheless, others have suggested that CTRP receptors may be distinct from adiponectin receptors [9], since phosphorylation of AMPK induced by CTRP5 remains unaffected by depletion of AdipoR1 or AdipoR2 [48]. Brain-specific angiogenesis inhibitor 3 (BAI3), a member of the cell-adhesion class of G-protein coupled receptors, has been proposed as an alternative receptor for a number of CTRPs including CTRP13 [49]. The orthologues of BAI3 in mice and humans are evolutionarily conserved and are widely distributed in various tissues and cells such as the brain [50], myoblasts [51], and some tumors [52]. Although one study reported an immune reaction to a BAI3 antibody in HUVECs [53], it is crucial to note that the antibody is not well characterized and lacks confirmation through other methods, such as Western blotting or gene expression analysis. Given this information, it is reasonable to speculate that CTRP13 may transmit its signal by binding to BAI3 in ECs. However, this hypothesis requires further investigation.

Adiponectin signal transduction, mediated via the adiponectin receptors 1 and 2 and T-cadherin [46,54], leads to APPL1-mediated activation of AMPK, Akt, p38 MAPK and PPAR-alpha [55,56]. On the other hand, CTRP13 activates the AMPK signaling pathway in various cell types, with Akt activation not observed universally [21]. Despite moderate differences in signal transduction, both adipokines share the common feature of contributing to the activation of AMPK in diverse cell types. In general, AMPK activation in vessels induces a robust vasodilation, primarily through upregulated eNOS phosphorylation and enhanced NO bioavailability. Additionally, it leads to alteration in endothelial metabolism, a reduction in oxidative stress, inflammation, and apoptosis, and the promotion of angiogenesis [57]. AMPK activation in vitro exerts a vasoprotective effect on EC senescence via its signaling involving mTOR, sirtuins, and eNOS [58]. The activation of AMPK induced by CTRP13 overexpression has also been demonstrated to alleviate palmitic acid-induced oxidative stress, inflammation, impaired angiogenesis, and endothelial dysfunction [59]. However, it is noteworthy that other vascular effects of CTRP13 have not been directly attributed to AMPK or even appear to occur independently of AMPK activation [15,16]. In addition, AMPK activation does not always confer beneficial effects on endothelial function. Constitutive activation of endothelial AMPK α1 does not confer vasoprotective effects in obese mice [60] and has even been demonstrated to promote vascular inflammation via the induction of COX-2 [61]. This suggests that prolonged activation of endothelial AMPK may even have detrimental effects. Moreover, it has also been shown that AMPK phosphorylates eNOS at the inhibitory site of Thr495, leading to decreased NO formation [62], and sustained activation of AMPK increases vascular inflammation [61].

Additionally, AMPK activation by various mediators has previously been demonstrated to inhibit EC proliferation [19] and regulate EC migration and differentiation [63]. Even adiponectin mediates differential effects on the endothelium: while globular adiponectin increased in vitro EC proliferation, migration, and angiogenesis, full-length adiponectin, the main adiponectin isoform in human plasma, had no impact on EC migration and angiogenesis [64]. In addition, proliferation and cell cycle progression of ECs induced by ox-LDL was shown to be inhibited by adiponectin [65]. AMPK activation also exerts anti-proliferative and anti-migratory actions in VSMCs by inhibiting cell cycle progression through increased Ser-15 phosphorylation, elevated p53 protein expression, and inhibition of Rb phosphorylation [66]. These observations highlight the complex and context-dependent role of AMPK in regulating vascular function. The data from our study align with this evidence, demonstrating that CTP13-mediated activation of AMPK resulted in reduced EC migration and proliferation through the inhibition of cell cycle progression. Similarly, the inhibitory paracrine effects of EC-derived CTRP13 on human fibroblast proliferation and migration have the potential to mediate a significant impact on the stability of coronary atherosclerotic plaques through the regulation of fibrotic ECM remodeling. While such anti-proliferative effects are generally protective in VSMCs with regard to arteriosclerosis development and progression, they may have deleterious effects through impaired endothelial barrier repair in vessels. On the other hand, abnormal proliferation of intimal ECs and of VSMCs contributes to intimal thickening of the aorta, which plays an important role in the development of atherosclerosis [67]. Accordingly, inhibition of EC proliferation and migration as induced by CTRP13 in our study per se is not necessarily detrimental. Despite the clear anti-proliferative effects of CTRP13, overall it may still be a vasoprotective adipokine through mechanisms involving NO production and inflammatory cell adhesion, as shown by others [15,16,18], or through its impact on EC differentiation and migration.

### Study Limitations

The above-mentioned discrepancy observed between CTRP13 gain-of-function and loss-of-function studies raises questions about how accurately the utilization of recombinant protein reflects the physiological effects of CTRP13, a concern that extends to the present study. Therefore, future investigations using KO models are warranted to test the hypothesis of whether or not CTRP13 indeed mediates vasoprotective effects and limits the development and progression of arteriosclerosis in vivo.

## 5. Conclusions

The role of CTRP13, which possesses significant vaso-modulatory properties, in cardiometabolic syndrome and atherosclerosis remains unclear, despite numerous reported cardiovascular effects. Our findings provide novel insight into the anti-proliferative effects of CTRP13 and potential underlying mechanisms of increased vascular CTRP13 expression. These insights should be taken into consideration when contemplating the use of CTRP13 as a vasoprotective and anti-atherosclerotic agent. Many aspects remain to be clarified concerning the signal transduction pathways involved, the receptors engaged, in vivo concentration levels, and the isoform-specific mechanisms of CTRP13. The observed differences between the CTRPs and adiponectin in mediating their effects on ECs may be attributed to variations in receptor/co-receptor affinity and the formation of distinct hetero- or homodimers. A comprehensive understanding of the molecular mechanisms of obesity linked to atherogenesis and the role of adipocytokines may help to facilitate the prevention and management of obesity-induced cardiovascular complications in the future.

## Figures and Tables

**Figure 1 cells-13-01291-f001:**
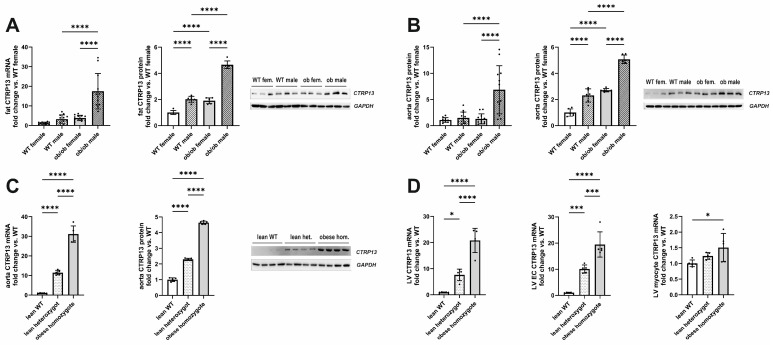
CTRP13 expression in obese and lean mice or rats. (**A**) CTRP13 mRNA and protein expression in adipose tissue of male and female wild-type (WT) or ob/ob mice. (**B**) CTRP13 mRNA and protein expression in aortic tissue of male and female WT or ob/ob mice. Representative CTRP13 western blots are shown. GAPDH served as loading control. Uncropped images with size markers are presented in Appendix A. Data are mean ± SEM from 6 to 12 animals per group and gender. **** *p* < 0.0001 vs. WT female. (**C**) CTRP13 mRNA and protein expression in aortic tissue of male obese ZDF (fa/fa) rats, lean heterozygous (Fa/fa) rats, and wild-type (Fa/Fa) rats. Representative CTRP13 western blots are shown. GAPDH served as loading control. Uncropped images with size markers are presented in Appendix A. (**D**) CTRP13 mRNA in the left ventricle (LV), LV ECs, and LV cardiomyocytes of male obese ZDF (fa/fa) rats, lean heterozygous (Fa/fa) rats, and wild-type (Fa/Fa) rats. Data are mean ± SEM from five animals per group and gender. * *p* < 0.05; *** *p* < 0.001, **** *p* < 0.0001 vs. wild-type (Fa/Fa) rats.

**Figure 2 cells-13-01291-f002:**
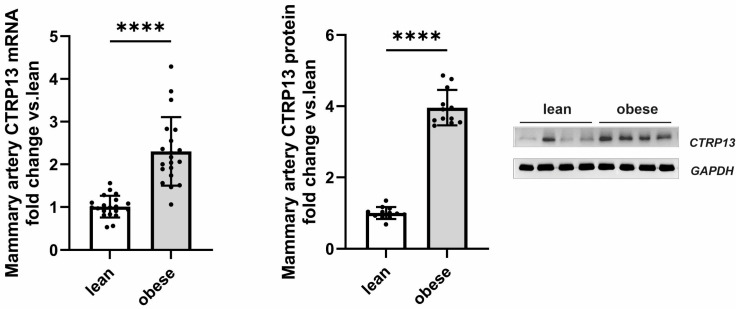
CTRP13 expression in the mammary artery of patients. CTRP13 mRNA and protein expression were analyzed in mammary arteries of lean (BMI 18.5–25 kg/m^2^) or obese (30–35 kg/m^2^) patients. Representative CTRP13 estern blots are shown. GAPDH served as loading control. Uncropped images with size markers are presented in Appendix A. Data are mean ± SEM from 12 to 18 patients per group. **** *p* < 0.0001 vs. lean patients.

**Figure 3 cells-13-01291-f003:**
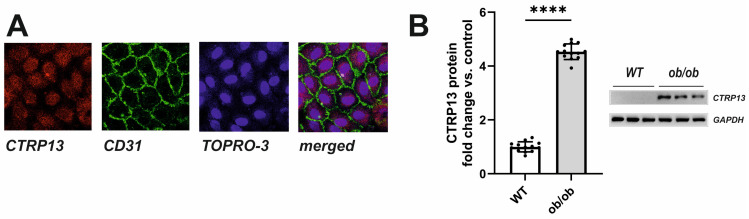
Intracellular localization and induction of CTRP13 in HUVECs. (**A**) Cells cultured on glass coverslips were stained with CTRP13 and CD31 followed by Cy3 or FITC-coupled secondary antibodies. The cell nuclei were stained with TO-PRO-3™ iodide (Ex./Em.642/661). (**B**) Cells were cultured in Opti-MEM™ supplemented with either 10% serum from wild-type (WT) or ob/ob mice for 48 h. Representative CTRP13 Western blots are shown. GAPDH served as loading control. Uncropped images with size markers are presented in Appendix A. Data are mean ± SEM from 4 independent experiments with 3–4 biological replicates each. **** *p* < 0.0001 vs. WT serum.

**Figure 4 cells-13-01291-f004:**
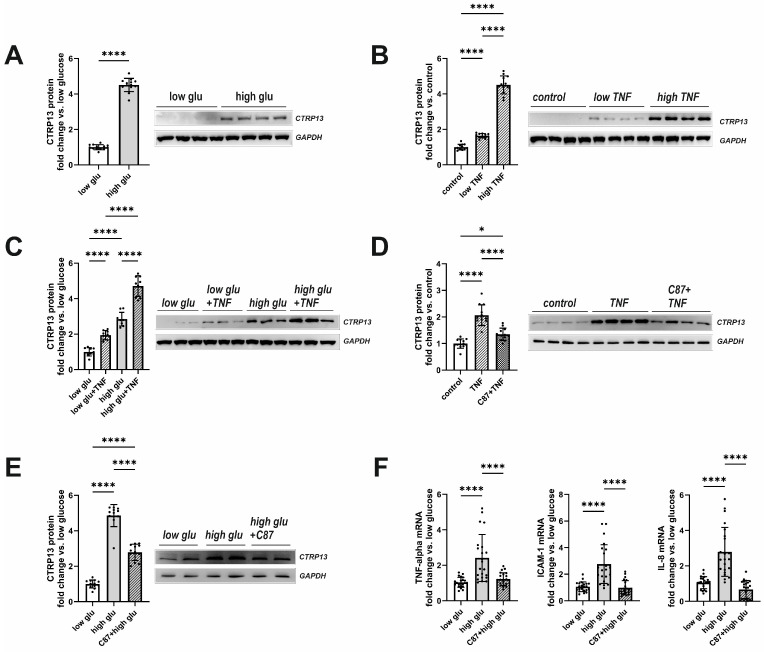
Impact of high glucose and TNF-alpha on CTRP13 protein expression in HUVECs. (**A**) HUVECs were cultured either under normal glucose (5 mM D-glucose, low glu) or high glucose (25 mM D-glucose, high glu) conditions for 24 h. (**B**) HUVECs were cultured under normal glucose conditions (5 mM D-glucose) and treated with 1 ng/mL (low TNF), 10 ng/mL (high TNF) TNF-alpha or buffer (control) for 24 h. (**C**) HUVECs were cultured under normal glucose conditions (5 mM D-glucose, low glu) or high glucose conditions (25 mM D-glucose, high glu) and treated ± 1 ng/mL TNF-alpha for 24 h. (**D**) HUVECs were cultured under normal glucose conditions (5 mM D-glucose, low glu), treated ± 1 ng/mL TNF-alpha for 24 h, and preincubated with the TNF-alpha-specific small-molecule inhibitor C87 (2 µM) for 60 min as indicated. (**E**) HUVECs were cultured under normal glucose conditions (5 mM D-glucose, low glu) or high glucose conditions (25 mM D-glucose, high glu) and treated ± C87 (2 µM) for 24 h. Representative CTRP13 Western blots are shown. GAPDH served as loading control. Uncropped images with size markers are presented in Appendix A. (**F**) Cells were treated as described in (**E**) and analyzed for the mRNA expression of TNF-alpha, ICAM-1, or IL-8 by qPCR. Data are mean ± SEM from 4 independent experiments with 3–5 biological replicates each. * *p* < 0.05, **** *p* < 0.0001 vs. normal glucose unless otherwise indicated.

**Figure 5 cells-13-01291-f005:**
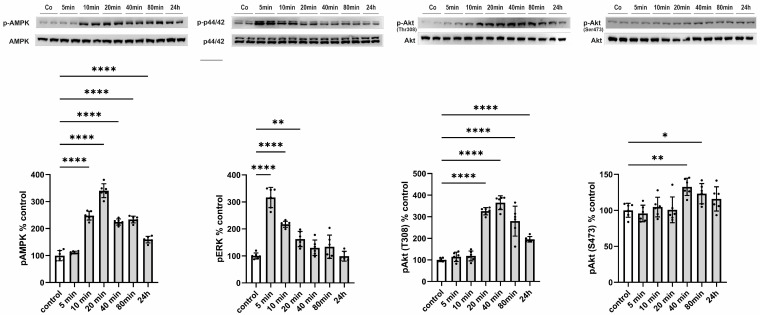
Time course of signaling activation in response to CTRP13 in HUVECs. Cells were treated with 4 µg/mL CTRP13 in Opti-MEM™ supplemented with 1% serum for the indicated duration. Western blots were performed to assess the phosphorylation of AMPK (Thr172), p44/42 MAPK (Thr202/Tyr204), and Akt (Thr308 or Ser473). Total AMPK, total p44/42 MAPK, or total Akt served as loading control. Uncropped images with size markers are shown in Appendix A. Data are mean ± SEM from five independent experiments with two biological replicates each. * *p* < 0.05, ** *p* < 0.01; **** *p* < 0.0001 vs. control.

**Figure 6 cells-13-01291-f006:**
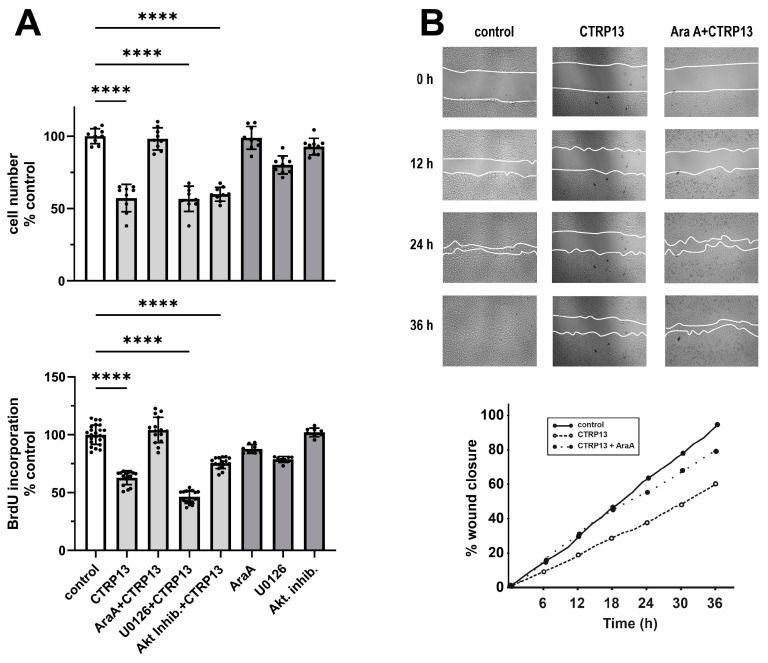
Impact of CTRP13 on EC proliferation and migration. (**A**) Cells were treated with 4 µg/mL CTRP13 in Opti-MEM™ supplemented with 1% serum for 24 h. Sixty min prior to CTRP13 treatment, HUVECs were incubated with the AMPK inhibitor AraA (500 µM), the p44/42 MAPKinase inhibitor UO126 (10 µM), or the Akt inhibitor VIII (0.1 µM). Subsequently, the cell numbers were counted (upper panel) or a BrdU assay was performed (lower panel). Data are mean ± SEM from 4 independent experiments with 2–5 biological replicates each. **** *p* < 0.0001 vs. control. (**B**) Cells were left untreated in Opti-MEM™ supplemented with 1% serum (control) or treated with 4 µg/mL CTRP13 for 24 h. Sixty min prior to CTRP13 treatment, HUVECs were incubated with the AMPK inhibitor AraA (500 µM) as indicated. To investigate the effect of CTRP13 on the migration of ECs, time-dependent closure of a cell-free gap with a width of 500 μm was monitored. Representative photomicrographs with a marked boundary of the cell-free gap are shown (**upper panel**) together with the quantification of 4 independent experiments with 1 biological replicate each (**lower panel**).

**Figure 7 cells-13-01291-f007:**
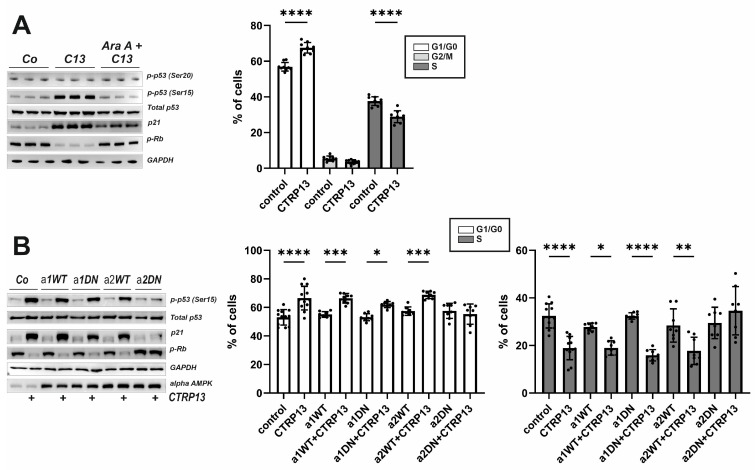
Impact of CTRP13 on cell cycle regulators and cell cycle progression. (**A**) Cells were left untreated in Opti-MEM™ supplemented with 1% serum or treated with 4 µg/mL CTRP13 for 24 h. Sixty min prior to CTRP13 treatment, HUVECs were incubated with the AMPK inhibitor AraA (500 µM) as indicated. Western blots were performed for phosphorylation of p53 (Ser15 or Ser20) or Rb and total protein expression of p53 and p21. Representative Western blots are shown (**left panel**). GAPDH served as loading control. Uncropped images with size markers are presented in Appendix A. Results from FACS-based cell cycle analyses in accordingly treated cells are shown in the right panel. (**B**) Adenoviral overexpression of wild-type (WT) or dominant negative (DN) alpha 1 and alpha 2 AMPK was performed in HUVECs. After 48 h, cells were left untreated in Opti-MEM™ supplemented with 1% serum or treated with 4 µg/mL CTRP13 for 24 h. Western blots and FACS-based cell cycle analyses were performed as described in (**A**). Data are mean ± SEM from 4 independent experiments with 2–3 biological replicates each. * *p* < 0.05, ** *p* < 0.01, *** *p* < 0.001, **** *p* < 0.0001 vs. control. Results regarding the impact of alpha AMPK isoforms in mediating the CTRP13 effects on the G2/M phase are presented in Appendix A.

**Figure 8 cells-13-01291-f008:**
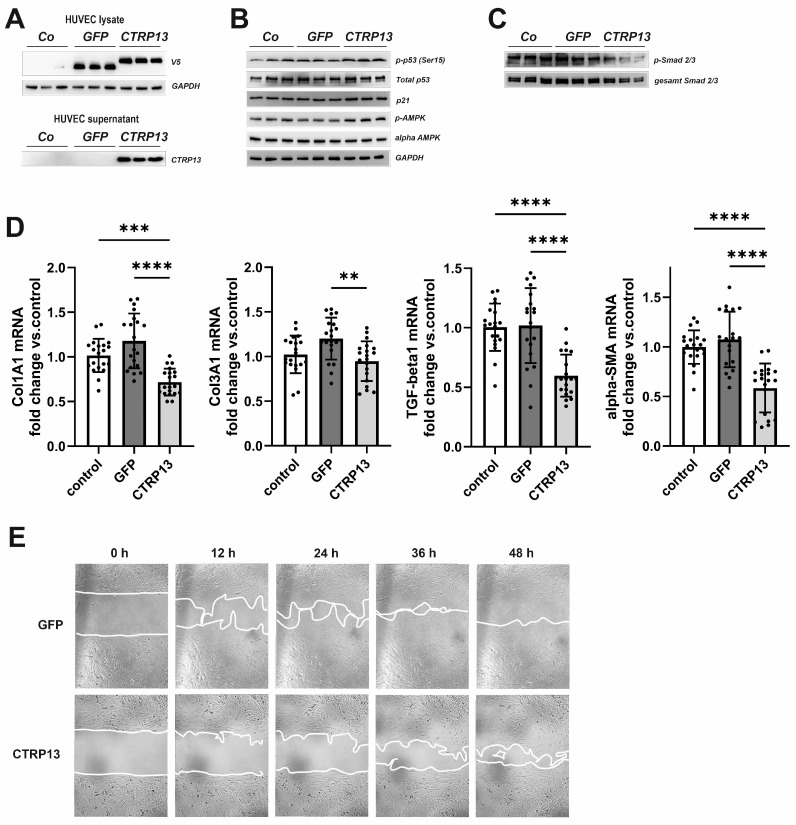
Impact of endothelial CTRP13 on human fibroblasts. Adenoviral overexpression of CTRP13 or GFP in HUVECs was performed. Conditioned medium from HUVECs overexpressing either CTRP13 or GFP or HUVECs treated with conditioned medium from 293A cells without overexpression was utilized to culture human fibroblasts in 50% (*v*/*v*) HUVECs medium in DMEM for 24 h (**B**,**C**) or as indicated (**E**). Overexpression of V5-tagged CTRP13 or GFP in HUVEC lysates (**upper panel**) or secretion of CTRP13 into cell culture medium (**lower panel**) was analyzed by Western blotting (**A**). GAPDH served as a loading control. (**B**) Western blots were performed for phosphorylation of p53 (Ser15), total protein expression of p53, p21, and alpha-AMPK and AMPK phosphorylation (Thr172). GAPDH served as loading control. (**C**) Western blots were performed for phosphorylation of Smad2 (Ser465/467)/Smad3 (Ser423/425). Representative Western blots are shown and uncropped images with size markers are presented in Appendix A. (**D**) Cells were analyzed for the mRNA expression of Col1A1, Col3A1, TGF-beta1, and alpha-SMA by qPCR. Data are mean ± SEM from 4 independent experiments with 4–5 biological replicates each. ** *p* < 0.01, *** *p* < 0.001, **** *p* < 0.0001 vs. control. (**E**) To investigate the effect of conditioned medium from HUVECs overexpressing either CTRP13 or GFP on the migration of fibroblasts, time-dependent closure of a cell-free gap with a width of 500 μm was monitored. Representative photomicrographs with a marked boundary of the cell-free gap are shown.

**Table 1 cells-13-01291-t001:** Mouse characteristics.

Parameter	Wild-Type (n = 12)	ob/ob (n = 12)
body weight (g)	31.5 ± 2.1	56.5 ± 4.8 ***
glucose (mmol/L)	7.1 ± 0.5	12.8 ± 0.9 *
insulin (µg/L)	1.1 ± 0.4	16.4 ± 0.5 ***
TNF-alpha (pg/mL)	20.5 ± 1.8	84.9 ± 2.1 ***

Data are mean ± SEM. * *p* < 0.05, *** *p* < 0.001 vs. wild-type mice.

**Table 2 cells-13-01291-t002:** Rat characteristics.

Parameter	Fa/Fa (n = 5)	Fa/fa (n = 5)	fa/fa (n = 5)
body weight (g)	331.6 ± 8.5	358.3 ± 10.2	465.5 ± 12.6 **
glucose (mmol/L)	5.5 ± 0.4	6.9 ± 0.6 *	16.3 ± 2.1 **
insulin (µg/L)	0.4 ± 0.3	1.7 ± 0.8 *	4.9 ± 0.6 **
TNF-alpha (pg/mL)	3.8 ± 0.4	9.7 ± 0.5 **	17.9 ± 1.5 ***

Data are mean ± SEM. * *p* < 0.05, ** *p* < 0.01, *** *p* < 0.001 vs. Fa/Fa rats.

**Table 3 cells-13-01291-t003:** Patient characteristics.

Parameter	Lean (n = 15)	Obese (n = 20)
BMI (kg/m^2^)	24.2 ± 0.4	33.5 ± 0.7 ***
age	69.0 ± 3.4	62.8 ± 3.2
male gender (n)	11	10
ejection fraction (%)	63.1 ± 2.2	59.9 ± 2.4
number of grafts (n)	3.6 ± 0.2	4.0 ± 0.3
cross-clamping time (min)	61.3 ± 4.6	59.7 ± 4.9
cardiopulm. Bypass (min)	102.9 ± 6.2	97.8 ± 7.7
preoperative CRP (mg/L)	5.0 ± 1.7	6.6 ± 2.0
preoperative leucocytes (10^9^/L)	7.4 ± 0.4	8.3 ± 0.4
preoperative HbA1c (%)	5.8 ± 0.1	6.0 ± 0.1
fasting glucose (mmol/L)	5.3 ± 0.9	7.5 ± 1.2 *
Beta-blocker (n)	10	10
ACE inhibitor (n)	8	7
ARB (n)	5	6
Anticoagulation (n)	15	15
Statin (n)	12	15
Diuretic (n)	6	5

Clinical parameters were registered before surgery. ARB: angiotensin II receptor blocker. Data are mean ± SEM. * *p* < 0.05, *** *p* < 0.001 vs. normal weight patients.

## Data Availability

The authors confirm that the data supporting the findings of this study are available within the article and its Appendix A. Further data that support the findings of this study are available on request from the corresponding author.

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
