# Peer review of "CTRP13-Mediated Effects on Endothelial Cell Function and Their Potential Role in Obesity"

_cells, 2024, doi:10.3390/cells13151291_

Round 1
Reviewer 1 Report
Comments and Suggestions for Authors
COMMENTS TO AUTHORS:
In this interesting manuscript, the authors investigated the CTRP13-mediated effects on endothelial cell function and their potential role in obesity. The authors elucidated how endothelial cell signaling is altered in the presence of CTRP13 in normal vs obese conditions and tried to clarify the previous misconceptions in the field. Overall, the study is mostly descriptive and well-articulated with supporting experiments. However, there are some limitations in the current manuscript that could be strengthened to improve rigor and overall impact.
1. The authors should perform the endothelial cell tube formation assay with and without obese condition and gain and loss of function of CTRP13 (siRNA to CTRP13) in the presence of TNFa.
2. Does Treatment with C87 have any impact on high glucose-induced CTRP13 levels? or does it only alter the TNFa-mediated CTRP13 induction?
3. Authors describe that CTRP13 may act via paracrine signaling; did they test the overexpression of CTRP13's effect on any neighboring smooth muscle cells or fibroblasts?
4. Western blot image labels should be highlighted appropriately with lines to differentiate the lanes with different groups or between treatment.
5. Authors should mention the number of n's in the respective figure legends, and the bar graph should overlap with the dot plot, which will help understand the sample distribution in the representative images.
6. Line 286 refers to Suppl. fig-4, not fig-5.
7. In lines 350 and 353 - (Tab. 1+2) should be written as (Tab. 1 and 2).
8. All the main text figures consist of western blot images, figure legends mentioning uncropped images, and "size markers," but no size markers are present in the figure. The legend needs to be corrected.
Comments on the Quality of English LanguageEasy to read and less percentage of plagiarism detected.
Reviewer 2 Report
Comments and Suggestions for Authors
This is a very interesting paper investigating CTRP13 regulation. The paper are new and will contribute to our understanding despite the fact that they do not finally give insight in its importance.
There might be remaining some mis-spellings ( at least I have seen one in line number 384 (must be "endothelial")
